# Muscle Loss after Chemoradiotherapy as a Biomarker of Distant Failures in Locally Advanced Cervical Cancer

**DOI:** 10.3390/cancers12030595

**Published:** 2020-03-05

**Authors:** Jie Lee, Jhen-Bin Lin, Meng-Hao Wu, Chih-Long Chang, Ya-Ting Jan, Yu-Jen Chen

**Affiliations:** 1Department of Radiation Oncology, MacKay Memorial Hospital, Taipei 104215, Taiwan; radionc@mmh.org.tw (M.-H.W.); oncoman@mmh.org.tw (Y.-J.C.); 2Department of Medicine, MacKay Medical College, Taipei 252005, Taiwan; clchang@mmc.edu.tw; 3Department of Radiation Oncology, Changhua Christian Hospital, Changhua 500209, Taiwan; 146894@cch.org.tw; 4Department of Obstetrics and Gynecology, MacKay Memorial Hospital, Taipei 104215, Taiwan; 5Department of Radiology, MacKay Memorial Hospital, Taipei 104215, Taiwan; gracilis.5319@mmh.org.tw

**Keywords:** cervical cancer, computed tomography, skeletal muscle loss, distant failures, chemoradiotherapy

## Abstract

This study aimed to evaluate whether computed tomography (CT)-based muscle measurement predicts distant failure in patients with locally advanced cervical cancer (LACC). Data from 278 patients with LACC who underwent chemoradiation therapy (CCRT) between 2004 and 2017 were analysed. Changes in the skeletal muscle index (SMI), skeletal muscle density, and total adipose tissue index during CCRT were calculated from CT images taken at the baseline and after CCRT. The predictive capability of CT-based muscle measurement for distant failure was evaluated using Cox proportional hazards regression, Harrell’s concordance index (C-index), and time-dependent receiver operating characteristic curves. SMI loss ≥ 5% was independently associated with worse distant recurrence-free survival (DRFS) (HR: 6.31, 95% CI: 3.18–12.53; *p* < 0.001). The addition of muscle change to clinical models, including International Federation of Gynaecology and Obstetrics (FIGO) stage, lymph nodes, pathology, and squamous cell carcinoma-antigen, achieved higher C-indices (0.824 vs. 0.756; *p* < 0.001). Models including muscle change had superior C-indices than those including weight change (0.824 vs. 0.758; *p* < 0.001). The area under the curve for predicting 3-year DRFS was the highest for the muscle-loss model (0.802, muscle-loss model; 0.635, clinical model; and 0.646, weight-loss model). Our study demonstrated that muscle loss after CCRT was independently associated with worse DRFS and that integrating muscle loss into models including classical prognostic factors improved the prediction of distant failure.

## 1. Introduction

Locally advanced cervical cancer (LACC) is a leading cause of morbidity and mortality in women worldwide [1]. The standard treatment for patients with LACC is cisplatin-based chemoradiation therapy (CCRT) that has favorable locoregional control. However, the main type of treatment failure after CCRT is distant failure [2,3,4,5,6]. Adjuvant chemotherapy after CCRT might improve survival outcomes [7], but its role remains controversial [8,9,10], and this may be attributed to the insufficiency of current classical risk stratification for high-risk patients [6]. The classical prognostic factors, such as International Federation of Gynaecology and Obstetrics (FIGO) stage, lymph nodes, histology, and serum squamous cell carcinoma-antigen (SCC-Ag), have been investigated in previous studies [2,3,4,5,6]. Hence, new biomarkers that can improve the prediction of distant failures are needed for patients with LACC.

Muscle loss after CCRT is an imaging biomarker of worse overall survival and progression-free survival in LACC (Figure 1) [11,12,13]. Body composition can be objectively evaluated using computed tomography (CT) images at the third lumbar vertebra (L3) level obtained for routine cancer care [14,15]. Furthermore, evaluating the body composition at a single time-point may not provide useful information and a longitudinal study plays an important role in understanding the impact of body composition on survival outcomes [16,17,18,19]. Cachexia phenotypes based on a specific CT-based body composition are also predictive for survival outcomes in cancer patients [20,21,22,23]. 

We hypothesized that muscle loss could be a potential biomarker of distant failure in LACC. Integrating CT-based muscle measurement could improve the prediction of survival outcomes in cancer patients [24,25]; however, the predictive value of muscle loss for distant failures is unknown in LACC. Therefore, in this study, we aimed to evaluate the predictive value of CT-based muscle measurement as a biomarker of distant failure in LACC.

## 2. Results

In total, 278 patients met the inclusion criteria (Appendix A). The final analysis included 556 CT scans from 278 patients. The clinical characteristics of the 278 patients are shown in Table 1. The median period from pre-treatment to post-treatment CT scans was 143 (interquartile range (IQR): 135–150) days. The median period from CCRT completion and post-treatment CT scans was 77 (IQR: 71–84) days.

### 2.1. Body Composition at the Baseline and Change after Treatment

The baseline body composition and changes after CCRT are summarized in Table 2 and Appendix A. Overall, patients significantly lost body mass index (BMI), skeletal muscle density (SMD), and total adipose tissue index (TATI) after CCRT; the skeletal muscle index (SMI) loss was marginally significant. Fifty-eight (20.9%) patients experienced weight loss of ≥5%, while 90 (32.4%), 124 (44.6%), and 125 (45.0%) patients experienced SMI loss, SMD loss, or TATI loss of ≥5%, respectively. SMI and SMD changes were not correlated with BMI change (Spearman’s ρ for SMI, 0.11; *p* = 0.06; ρ for SMD, 0.10; *p* = 0.10; Figure 2A,B). TATI changes were weakly correlated with BMI change (Spearman’s ρ for TATI, 0.35; *p* < 0.001; Figure 2C). SMI changes were moderately correlated with SMD change (Spearman’s ρ for SMD, 0.54; *p* < 0.001; Figure 2D). TATI changes were weakly correlated with SMI and SMD changes (Spearman’s ρ for SMI, 0.19; *p* = 0.001; ρ for SMD, 0.12; *p* = 0.06; Figure 2E,F).

The cut-off values for body compositions were as follows: sarcopenia: SMI < 36.3 cm^2^/m^2^; myosteatosis: SMD < 30.7 HU; and low TATI: TATI < 112.2 cm^2^/m^2^. Pre-treatment BMI, SMI, SMD, and TATI were similar for the SMI loss and SMI maintained groups (Figure 3; Appendix A). The frequency of sarcopenia before CCRT was similar for SMI loss and SMI maintained groups (28.9% vs. 35.1%; *p* = 0.30). After CCRT, the frequency of sarcopenia was significantly higher in the SMI loss group than in the SMI maintained group (48.9% vs. 28.2%; *p* = 0.001). Pre-treatment myosteatosis was present in 31 (34.4%) patients in the SMI loss group compared with 61 (32.4%) patients in the SMI maintained group (*p* = 0.74). Pre-treatment low TATI was present in 57 (63.3%) patients in the SMI loss group compared with 129 (68.6%) patients in the SMI maintained group (*p* = 0.38).

Adenocarcinoma was associated with SMI loss (Table 1). Compared with patients with squamous cell carcinoma (SCC), patients with adenocarcinoma lost more SMI (−5.0% vs. −0.5%; *p* = 0.01) and marginally more SMD (−6.7% vs. −2.4%; *p* = 0.06) (Appendix A). Demographic characteristics such as FIGO stage, pelvic lymph nodes (PLNs), SCC-antigen (SCC-Ag), and treatments were not significantly different between groups (Table 1).

### 2.2. Body Composition and Distant Failures

The median follow-up duration was 52.4 months (IQR, 29.9–93.1 months). Patterns of failures according to body composition groups are shown in Appendix A. Distant failures were observed significantly more often in patients with SMI loss or SMD loss groups; no difference was observed in the incidence of distant failure between the weight loss and weight maintained groups. There were no significant differences in pelvic failures alone according to the body composition groups. The median interval from post-treatment CT scans to distant failure diagnosis was 8.4 months (IQR, 5.1–16.4 months) in the SMI loss group and 19.3 months (IQR, 6.8–45.7 months) in the SMI maintained group (*p* = 0.04). The median interval from post-treatment CT scans to distant failure diagnosis was 9.4 months (IQR, 5.6–26.2 months) in the SMD loss group and 10.5 months (IQR, 3.5–19.8 months) in the SMD maintained group (*p* = 0.78). The 3-year distant recurrence-free survival (DRFS) rates were 79.7% and 82.3% in the weight loss and weight maintained groups, respectively (*p* = 0.65; Figure 4A); 56.3% and 94.0% in the SMI loss and SMI maintained groups, respectively (*p* < 0.001; Figure 4B); and 74.6% and 88.4% in the SMD loss and SMD maintained groups, respectively (*p* = 0.002; Figure 4C). Patients with TATI loss had marginally lower 3-year DRFS rates (Figure 4D). Stratifying the patients into three categories, including “No SMI loss”, “Only SMI loss”, and “SMI and TATI losses”, the 3-year DRFS were 93.3%, 52.6%, and 59.2%, respectively (*p* < 0.001; Figure 5).

In univariable analysis, SMI, SMD, and TATI change; FIGO stage; PLN status; pathology; and SCC-Ag were 3-year DRFS predictors (Appendix A). Pre-treatment sarcopenia, myosteatosis, and low TATI were not associated with 3-year DRFS. In multivariable analysis (Table 3), SMI loss was independently associated with worse 3-year DRFS (hazard ratio (HR): 6.31, 95% confidence (CI): 3.18–12.53; *p* < 0.001). Weight, SMD, and TATI change were not associated with 3-year DRFS. In the subgroup analysis including only SCC (*n* = 246), SMI loss remained an independent prognostic factor for 3-year DRFS (HR: 5.86, 95% CI: 2.85–12.08; *p* < 0.001) (Appendix A).

### 2.3. Comparison of Clinical, Weight-Loss, and Muscle-Loss Models

To evaluate the benefit of incorporating CT-based muscle measurements for predicting distant failures, we constructed DRFS models incorporating muscle or weight change and other clinical prognostic variables and determined the C-index, receiver operating characteristic (ROC) curves, and area under the curve (AUC). The clinical model included the FIGO stage, PLN status, pathology, and SCC-Ag; the weight-loss model added weight change to the clinical model; and the muscle-loss model added muscle change to the clinical model. The C-indexes of the constructed models showed similar indexes for the clinical model (0.756 [95% CI: 0.679–0.832]) and the weight-loss model (0.758 [95% CI: 0.680–0.836]), *p* < 0.06, while that of the muscle-loss model (0.824 [95% CI: 0.748–0.900]) was greater than that of the clinical model (*p* < 0.001) and weight-loss model (*p* < 0.001). Figure 6 shows the time-dependent ROC curves for 3-year DRFS for the three models. The muscle-loss model achieved the highest area under the curve for 3-year DRFS prediction (0.635, 0.646, and 0.802 for the clinical model, the weight-loss model, and the muscle-loss model, respectively).

## 3. Discussion

To the best of our knowledge, this is the first study to evaluate the predictive value of CT-based muscle measurement as a biomarker of distant failure in LACC. We found that muscle loss after CCRT was independently associated with distant failures. Patients with muscle loss had significantly shorter time to distant failures than patients with no muscle loss. SMI and SMD changes were not correlated with BMI changes, suggesting that muscle loss was occult and occurred independently of BMI change. Integrating CT-based muscle measurement into models that include classical prognostic factors may improve the prediction for distant failures.

The mechanism for the association of muscle loss with distant failure is unknown. The possible explanation of our findings might be cancer-associated muscle loss after CCRT [26,27,28]. The circulating tumor cells (CTCs) in LACC may not be completely eradicated by CCRT [3,4,5,6,29]. Elevated CTCs levels are also associated with poor disease-free survival in LACC [29], implying that residual CTCs after CCRT might contribute to progressive muscle loss before resulting in gross detectable distant failures [30,31]. In the current study, we found that patients with muscle loss had a significantly shorter time to distant failure and worse DRFS. Therefore, patients with residual CTC may experience rapid muscle loss after CCRT. However, the association between CTC and muscle loss could not be evaluated in this study. The CTC analysis is not the current standard of care in staging and follow-up in LACC [9]. In addition, we also found that the changes in body composition were different between SCC and adenocarcinoma; it implied that the protein synthesis rate may differ between SCC and adenocarcinoma, and may affect whole-body protein metabolism in humans [28]. The nutritional status may also deteriorate during CCRT and affect the clinical outcome in LACC patients [13]. Although we found that muscle loss after CCRT was a predictor of distant failures, our hypothesis and findings should be validated in future studies.

BMI is not a precise measure of body composition because patients with a similar BMI can have different body composition phenotypes [32,33]. Cancer cachexia is characterized by progressive muscle loss with or without a loss of fat mass that cannot be fully reversed with nutritional support [34]. However, the current definition of cachexia is based on weight loss, without considering the changes in body composition [34]. In a previous study, it was reported that CT-based body composition analysis detected muscle or fat loss of >5% in 81% of patients, while the traditional definition of weight loss of >5% was observed in only 56.6% of patients [21]. If the traditional definition of weight loss of >5% was applied to patients with muscle or fat loss of >5%, this would lead to missing nearly 30% of these patients, as they would have been reported as having developed cachexia in their study. Moreover, three distinct cachexia phenotypes were suggested based on muscle and fat changes and their impact on outcomes in pancreatic cancer patients [21]. In this study, we found that SMI and SMD changes were not correlated with BMI change, and that the TATI change was weakly correlated, although significantly, with BMI change. These findings suggested that the changes in weight during cancer therapy could not have represented muscle change, but might likely represent TATI change. Applying the traditional weight loss criterion to patients with SMI loss would have classified 24/90 patients as having developed cachexia; this would have missed 66/90 or 73.3% of patients with SMI loss. Furthermore, we found that muscle change, but not fat change, was associated with distant failures in LACC patients treated with CCRT. Differences in the effect of fat on outcomes might be due to differences in cancer types, treatments, or ethnicity. Weight loss based on the current definition of cachexia was not predictive of distant failures in LACC. When predicting distant failures, the muscle-loss model also significantly improved the prediction of distant failures compared with either the clinical or weight-loss models. Hence, defining cachexia phenotypes based on the CT-based body composition measurement rather than the weight alone, may provide a more precise definition of cachexia phenotypes. Integrating the skeletal muscle measurement into the classical prognostic models could also help improve the prediction of distant failures in cancer patients.

A longitudinal study of body composition can provide a comprehensive understanding of body composition changes during cancer therapy and their impact on outcomes in cancer patients [16,17,18,19]. Skeletal muscle loss after CCRT was associated with worse survival outcomes in LACC [11,12,13]. Previous studies also reported that pre-treatment sarcopenia was not a predictor of outcomes in LACC [11,12]. In the present study, the pre-treatment BMI, SMI, SMD, and TATI were not significantly different between the SMI loss and SMI maintained groups. After CCRT, patients experienced a significant loss in BMI, SMD, and TATI; the SMI loss was marginally significant. Patients with SMI loss also had significantly higher SMD and TATI loss. These findings suggest that evaluating body composition parameters at a single specific time point may not help in predicting distant failure.

There were some limitations of our study. First, the retrospective nature of this study limits the longitudinal assessment of nutritional status, muscle strength, physical performance (e.g., gait speed), or systemic inflammatory markers (e.g., modified Glasgow prognostic score) during the course of CCRT [13]. Second, this study could not infer causality between the body composition parameters and distant failures. Third, CTCs were not evaluated in this study.

## 4. Materials and Methods 

### 4.1. Patients

Our Institutional Review Board approved the study and waived the requirement for informed consent owing to the retrospective and observational nature of this study (16MMHIS060e and 19MMHIS031e). Patients at our institution with FIGO stage IB2–IVA cervical cancer treated with definitive radiotherapy or CCRT between 2004 and 2017 were reviewed. The inclusion criteria were as follows: (i) routine abdominal CT performed before and after CCRT completion, (ii) sufficient quality of both CT scans to perform accurate measurements of tissue area, and (iii) sufficient relevant clinical data from the patient’s medical records.

All patients received intensity-modulated radiotherapy comprising 6–9 coplanar fields using 6- or 10-MV photons. The pelvis was the standard radiation field, with a prescribed dose of 45.0–50.4 Gy. Para-aortic irradiation was considered for patients with positive pelvic PLNs or FIGO stage III–IVA disease [3,4]. The dose to the involved PLNs was boosted up to 60 Gy. The prescribed dose for brachytherapy was 5 Gy to point A for 6 sessions. Chemotherapy comprised weekly cisplatin (40 mg/m^2^) administered concurrently with radiotherapy. The treatment duration of CCRT was 7–9 weeks, with a routine pre-treatment CT image obtained before CCRT and a post-treatment CT image obtained within 3 months of CCRT completion.

### 4.2. Computed Tomography-Based Body Composition Analysis

Pre- and post-treatment CT images were retrieved for analysis. Body weight and height were obtained from medical records within 2 weeks of the initial and follow-up CT scans. In our institution, routine abdominal and pelvic CT images were obtained from female patients after the intravenous administration of a single uniphasic bolus dose (80–100 mL) of iohexol 300 (Omnipaque 300, GE Healthcare) or iopromide 300 (Ultravist 300, Bayer HealthCare) using a power injector at 2 mL/s. The portal-venous phase was obtained with a pitch between 1.0 and 1.5 and a fixed delay of 70 s since the administration of the contrast material before the contrast medium was excreted into the bladder. The information in the CT image parameters included contrast-enhanced, 5-mm slice thickness, 120 kVp, and approximately 290 mA. 

Two consecutive transverse CT images extending from L3 to the iliac crest were analysed using the Varian Eclipse software (Varian Medical Systems Inc., Palo Alto, CA, USA) [24]. Predetermined Hounsfield unit (HU) thresholds were −29 to +150 HU for skeletal muscle, −50 to −150 HU for visceral adipose tissue, and −30 to −190 HU for subcutaneous and intermuscular adipose tissue [14,15,20]. The cross-sectional areas (cm^2^) of skeletal muscle (including the psoas, paraspinal, transversus abdominis, rectus abdominis, and internal and external oblique muscles) and adipose tissues were calculated. The skeletal muscle density (SMD) was defined as the average radiation attenuation of skeletal muscle area in HUs [25]. The total adipose tissue (TAT) area was the sum of visceral, intermuscular, and subcutaneous adipose tissue areas. A researcher who was blinded to the patients’ information and outcomes measured the body composition parameters. An average of the measured body composition parameters from two consecutive images was obtained. In a sample of 90 patients randomly selected from this cohort, intraobserver coefficients of variation were 1.2%, 1.0%, and 1.5% for the skeletal muscle area, SMD, and TAT area, respectively, which is consistent with a previous paper in the literature [20]. The cross-sectional areas of skeletal muscle and TAT were normalized for the patients’ heights to calculate the indexes (cm^2^/m^2^) for skeletal muscle (SMI) and TAT (TATI) [20,25].

Because body composition varies greatly across regions, ethnicities, and cancer types, we defined our own cut-off values for defining sarcopenia, myosteatosis, and low TATI as in previous studies with similar population sizes. Cut-off values were defined as the lowest tertile for SMI and SMD and the highest tertile for TATI [35,36,37]. The post-treatment body composition change was the difference between the pre- and post-treatment CT images. The treatment duration of CCRT was 7–9 weeks, with post-treatment CT images obtained within 3 months of CCRT completion [9]. To account for variation in the scan interval duration, body composition changes were calculated as change per 150 days to provide a standardized unit for comparisons of patients. As per the current definition of cachexia [34], patients with a reduction in weight, SMI, SMD, or TATI ≥ 5% were classified as having ‘loss’ [21].

### 4.3. Statistical Analysis

Continuous and categorical data were presented as the mean ± standard deviation or median and IQR and numbers (%), respectively. The distributions of patient characteristics were compared using the Chi-square test and independent *t*-test for categorical and continuous variables, respectively. Paired *t*-tests and the Wilcoxon’s signed-rank test were used to assess the changes in body composition. Spearman’s correlation coefficient was used to assess the relationships between body composition parameters (BMI, SMI, SMD, and TATI).

Distant failure was defined as recurrence in non-regional lymph nodes (mediastinal or supraclavicular region) or visceral metastasis. Pelvic failure was defined as recurrence in the cervix, adjacent pelvic organs (e.g., parametrium, bladder, and vagina), or PLNs. Failure was recorded on the basis of clinical examination and imaging findings with pathology proven where possible. The recurrence-free survival was defined as the time from diagnosis to recurrence or death from any cause. The Kaplan–Meier method with log-rank tests was used to construct the survival curves. To evaluate the benefit of incorporating muscle measurements and other clinical prognostic parameters for predicting distant failures, we constructed Cox proportional hazards regression models. Univariable Cox proportional hazards regression analysis was performed to assess the prognostic factors for recurrence-free survival. All variables with *p* < 0.10 in the univariable analysis or with known clinical relevance were included in the multivariable Cox proportional hazards regression analysis.

Multivariable Cox proportional hazards regression analysis was used to develop multivariable models, including a “clinical model” (which included clinical prognostic parameters), a “weight-loss model” (adding weight change to the “clinical model”), and a “muscle-loss model” (adding SMI and SMD change to the “clinical model”). Clinical parameters considered for their potential prognostic ability in the LACC included the FIGO stage (IB-II vs. III-IVA), PLN involvement (negative vs. positive), pathology (SCC vs. adenocarcinoma), and SCC-Ag (continuous variable) [2,3,4,5,6].

The proportional hazards assumption was evaluated using the test of proportionality based on the Schoenefeld residuals. We developed 1000 bootstrap replications that were used as internal ation subsets to estimate the bias-corrected Harrell’s concordance index (C-index) and determine the discrimination ability. The z-score test (Package “compareC” in R) was used to test the difference between two C-indexes [38]. To summarize the predictive accuracy of the 3-year recurrence-free survival, time-dependent receiver operating characteristic curves and area under the curve functions were used [39,40]. The data were analysed using IBM SPSS software (version 21.0; IBM Corp., Armonk, NY, USA) and R software, version 3.5.1. (http://www.r-project.org), including the rms, compareC, and timeROC packages [38,39,40,41]. A *p*-value < 0.05 was considered statistically significant.

## 5. Conclusions

In conclusion, this study demonstrated that integrating CT-based muscle measurements in classical prognostic models could improve distant failure prediction in LACC. The changes in weight were not correlated with the changes in muscle. Therefore, CT-based muscle measurement should be considered to detect early muscle loss in clinical practice. Our findings suggest that CT-based muscle measurements as patient-specific biomarkers may improve outcome predictions and potentially guide cancer therapy or supportive care. Future studies are needed to validate the clinical utility of muscle measurements in cancer patients.

## Figures and Tables

**Figure 1 cancers-12-00595-f001:**
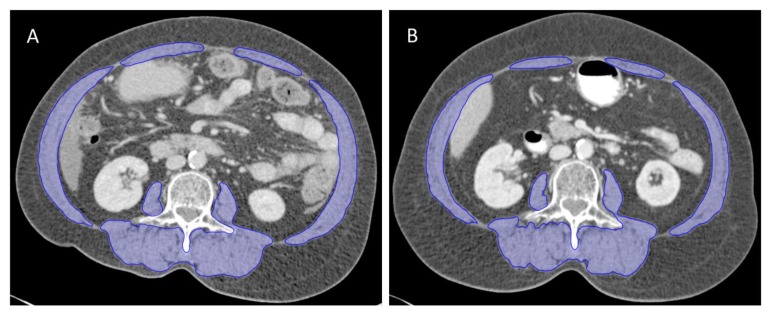
The change of skeletal muscle area (outlined in blue) on computed tomography (CT) images at the L3 vertebral level at the baseline (**A**) and after chemoradiation therapy (**B**) from one patient. Images were taken 5 months apart. At the time point of pre-treatment CT, the body mass index (BMI) and skeletal muscle index (SMI) were 24.1 kg/m^2^ and 51.8 cm^2^/m^2^, respectively. At the time point of post-treatment CT, the BMI and SMI were 25.4 kg/m^2^ and 47.3 cm^2^/m^2^, respectively.

**Figure 2 cancers-12-00595-f002:**
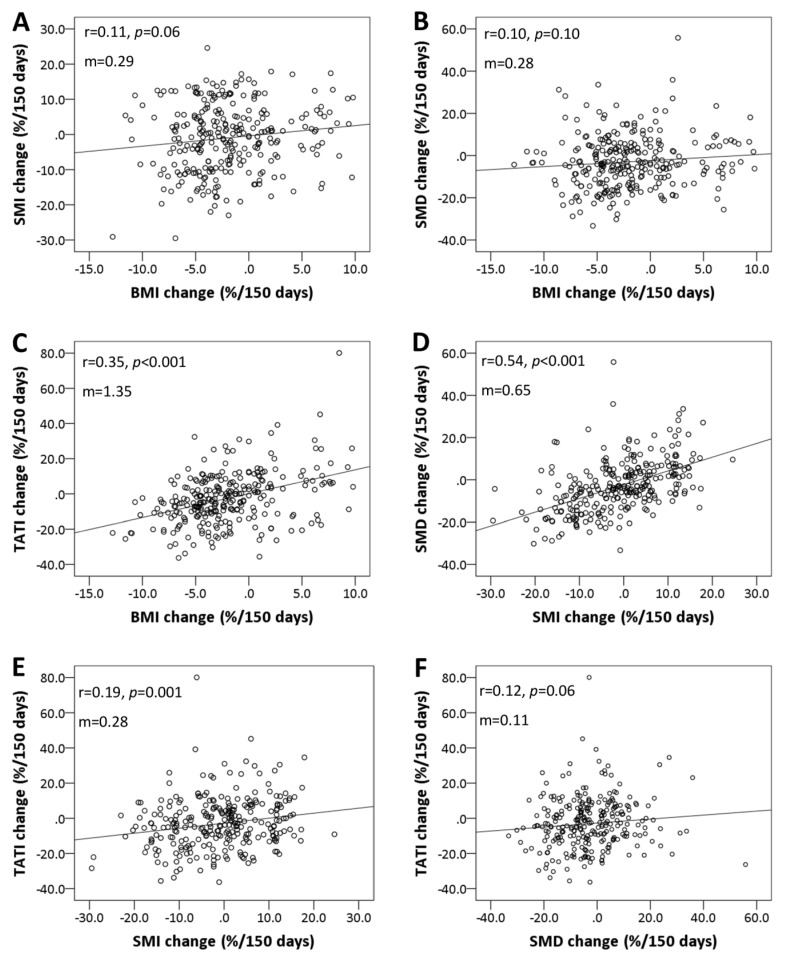
Scatter plots showing correlations between the changes in BMI, SMI, skeletal muscle density (SMD), and total adipose tissue index (TATI): (**A**) SMI vs. BMI, (**B**) SMD vs. BMI, (**C**) TATI vs. BMI, (**D**) SMD vs. SMI, (**E**) TATI vs. SMI, and (**F**) TATI vs. SMD. Spearman’s rank correlation coefficient (rho) was used to assess correlations between body composition parameters. Slopes (m) for the correlations are shown on each graph.

**Figure 3 cancers-12-00595-f003:**
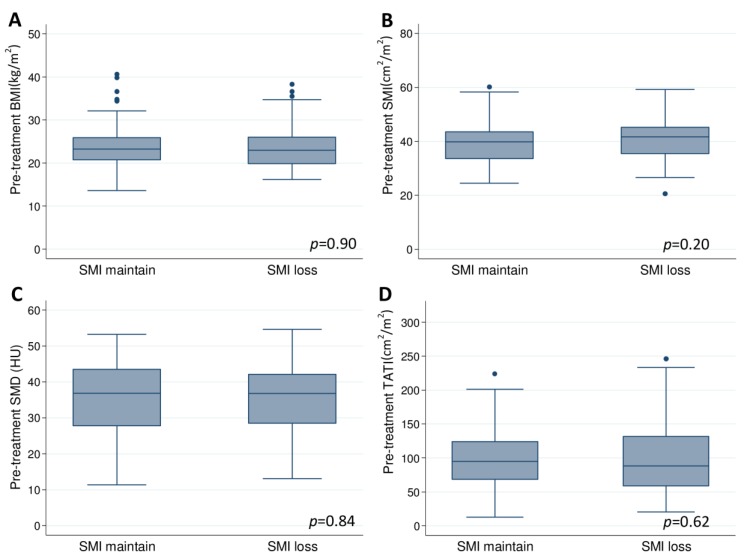
Pre-treatment (**A**) BMI, (**B**) SMI, (**C**) SMD, and (**D**) TATI according to SMI change groups.

**Figure 4 cancers-12-00595-f004:**
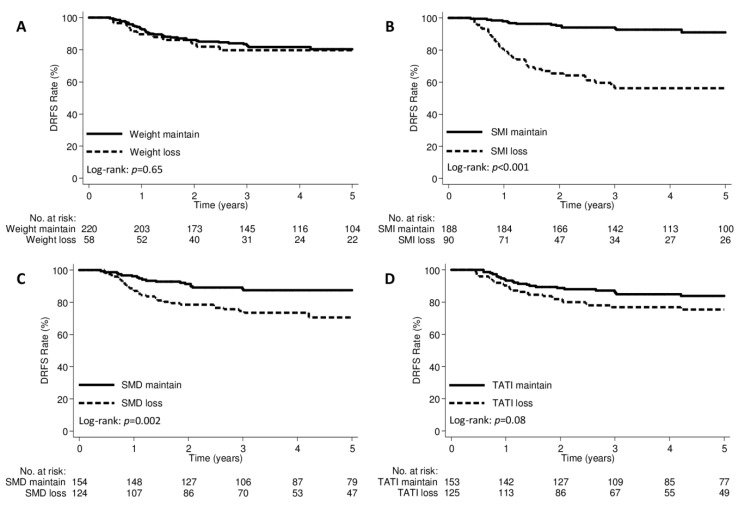
Kaplan–Meier curve demonstrating distant recurrence-free survival according to (**A**) weight change, (**B**) SMI change, (**C**) SMD change, and (**D**) TATI change groups. DRFS, distant recurrence-free survival; SMD, skeletal muscle density; SMI, skeletal muscle index; TATI, total adipose tissue index.

**Figure 5 cancers-12-00595-f005:**
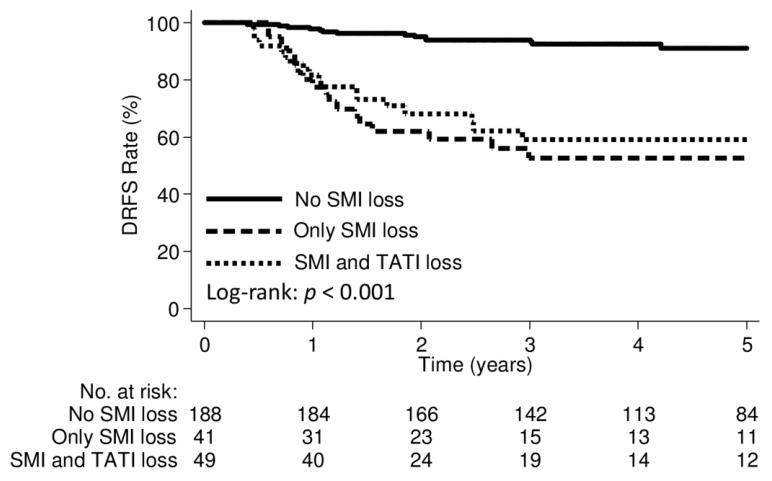
Kaplan–Meier curve demonstrating distant recurrence-free survival in patients with “No SMI loss”, “Only SMI loss”, and “SMI and TATI losses”.

**Figure 6 cancers-12-00595-f006:**
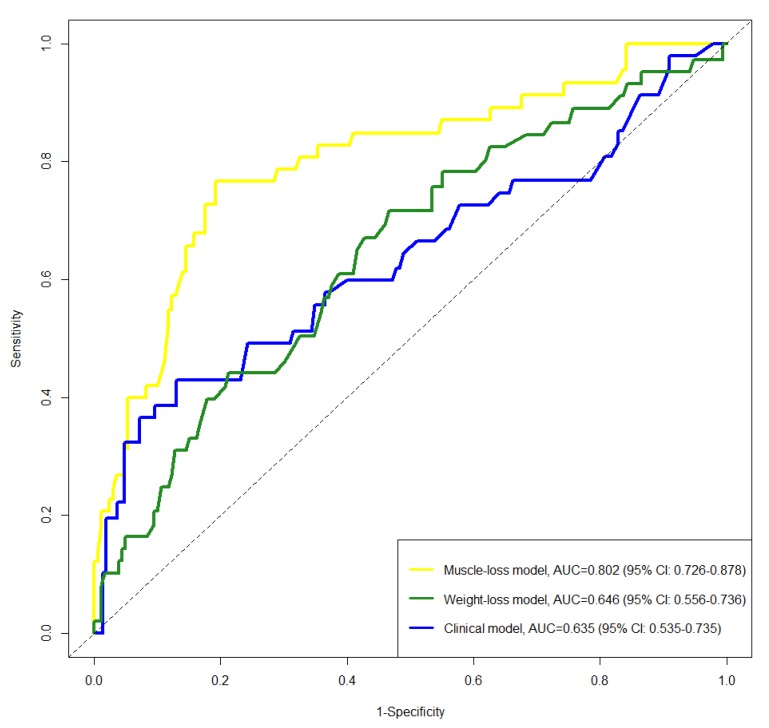
Receiver operating characteristic curves for 3-year distant recurrence-free survival in overall patients (*n* = 278).

**Table 1 cancers-12-00595-t001:** Characteristics of the included patients.

Characteristics	Overall (*n* = 278)	SMI loss (*n* = 90)	SMI Maintained (*n* = 188)	*p*-Value
**Age (years), median (IQR)**	62 (53–73)	64 (54–75)	62 (52–71)	0.11
**ECOG performance status**				0.08
0	253 (91.0)	78 (86.7)	175 (93.1)	
1	25 (9.0)	12 (13.3)	13 (6.9)	
**FIGO stage**				0.28
IB-II	203 (73.0)	62 (68.9)	141 (75.0)	
III-IVA	75 (27.0)	28 (31.1)	47 (25.0)	
**Pathology**				0.002
Squamous cell carcinoma	246 (88.5)	72 (80.0)	174 (92.6)	
Adenocarcinoma	32 (11.5)	18 (20.0)	14 (7.4)	
**P** **elvic lymph node**				0.80
Positive	136 (48.9)	45 (50.0)	91 (48.4)	
Negative	142 (51.1)	45 (50.0)	97 (51.6)	
**SCC-Ag level, median (IQR)**	7.6 (3.3–15.0)	8.6 (2.8–20.4)	6.9 (3.3–14.3)	0.29
**Radiation field**				0.09
Extended-field radiotherapy	147 (52.9)	41 (45.6)	106 (56.4)	
Pelvic radiotherapy	131 (47.1)	49 (54.4)	82 (43.6)	
**Chemotherapy**				0.07
Yes	243 (87.4)	74 (82.2)	169 (89.9)	
No	35 (12.6)	16 (17.8)	19 (10.1)	
**Chemotherapy cycles**	*n* = 243	*n* = 74	*n* = 169	0.78
5–6	181 (74.5)	56 (75.7)	125 (74.0)	
1–4	62 (25.5)	18 (24.3)	44 (26.0)	
**Overall treatment duration (day), median (IQR)**	58 (54–61)	59 (52–61)	58 (54–61)	0.89
**Median (IQR) duration between CT scans, days**	143 (135–150)	141 (133–149)	143 (135–151)	0.38

Data are the mean ± standard deviation or *n* (%).

**Table 2 cancers-12-00595-t002:** Change of body composition parameters after CCRT.

Variable	First CT Scan	Second CT Scan	Relative Change Per 150 Days (%)		
	Mean ± SD	Mean ± SD	Mean	95% CI	*p*-Value
BMI (kg/m^2^)	23.4 ± 4.3	23.0 ± 4.3	−1.9	−2.4 to −1.4	<0.001
SMI (cm^2^/m^2^)	39.8 ± 7.3	39.3 ± 7.6	−1.0	−2.1 to 0.2	0.09
SMD (HU)	35.6 ± 9.5	34.5 ± 9.6	−2.9	−4.4 to −1.5	<0.001
TATI (cm^2^/m^2^)	97.7 ± 44.5	93.7 ± 41.8	−3.0	−4.9 to −1.2	0.001

**Table 3 cancers-12-00595-t003:** Multivariable Cox proportional hazards model for 3-year distant recurrence-free survival.

Variable	Clinical Model		Weight-Loss Model		Muscle-Loss Model	
	HR (95% CI)	*p*-Value	HR (95% CI)	*p*-Value	HR (95% CI)	*p*-Value
**FIGO stage**						
IB-II	Reference		Reference		Reference	
III-IVA	2.30 (1.21–4.37)	0.01	2.36 (1.24–4.47)	0.01	1.98 (1.05–3.74)	0.04
**PLNs involvement**						
Negative	Reference		Reference		Reference	
Positive	1.75 (0.87–3.50)	0.12	1.81 (0.90–3.63)	0.10	2.31 (1.24–4.30)	0.01
**Pathology**						
SCC	Reference		Reference		Reference	
Adenocarcinoma	4.43 (2.30–8.53)	<0.001	4.63 (2.40–8.94)	<0.001	3.03 (1.54–5.95)	0.001
**SCC-Ag**	1.02 (1.01–1.02)	<0.001	1.02 (1.01–1.02)	<0.001	1.02 (1.01–1.02)	0.001
**Weight change**						
Weight maintain	-	-	Reference		-	-
Weight loss	-	-	1.51 (0.76–3.02)	0.24	-	-
**SMI change**						
SMI maintain	-	-	-	-	Reference	
SMI loss	-	-	-	-	6.31 (3.18–12.53)	<0.001
**SMD change**						
SMD maintain	-	-	-	-	Reference	
SMD loss	-	-	-	-	1.00 (0.53–1.87)	0.99
**TATI change**						
TATI maintain	-	-	-	-	Reference	
TATI loss	-	-	-	-	1.32 (0.73–2.38)	0.36

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
