# Peer review of "Muscle Loss after Chemoradiotherapy as a Biomarker of Distant Failures in Locally Advanced Cervical Cancer"

_cancers, 2020, doi:10.3390/cancers12030595_

Round 1

Reviewer 1 Report

In the submitted study on „Muscle loss after chemoradiotherapy as a biomarker of distant failures in locally advanced cervical cancer” the authors analyzed the potential of SMI and SMD measurements as suitable markers to predict distant failures after CCRT of patients with locally advanced cervical cancer.

The authors systematically and thoroughly analyze BMI, SMI, SMD and TATI for the study cohort and provide comprehensive detail information on their observations supporting the claims made in the manuscript.

The authors discuss their findings critically, considering different aspects of causality and correlation of their observations in the discussion (MS lines 162-180). They explain the overall setup of the study and the selection of the cohort and also point out that the pre-treatment BMI, SMI, SMD and TATI were not significantly different between SMI loss and SMI maintained groups.

Further, they point out the limitations of their study and suggest possible next steps.

Although the authors mention another study analyzing SMI changes in comparison to weight loss measurements (MS line187-189) it does not become sufficiently clear whether this is the same paper they are referring to in Ref 20 (MS line 190) or whether this is another study. Also, the authors do not emphasize the comparison to their study sufficiently, they could point out more clearly that the overall results of both studies are comparable and further strengthen the importance to include SMI measurements in clinical predictions. This should ideally already be mentioned in the introduction where they in line 64 and 65 just refer to the current state of knowledge in the context of LACC. There the authors should refer to their study in a broader context. Are SMI and SMD measurements performed or described to increase predictions in the context of any other form of cancer?

Further, I recommend that the authors consider providing more visual display formats for the information provided in the tables and the text. It would be easier for the reader if some comparisons would be plotted as bar graphs in addition to the tables for example in line 143-145 listing the C-indexes.

Same advice holds true for Table 1 and 2, this should be plotted as bar graphs or similar in addition to the data listed in the tables (the tables could still be provided as supplemental information). Table 1 and 2 contain a lot of information and plotting this data is essential in order to present this data in the best comprehensible format.

Overall, the authors could convincingly show that muscle loss after CCRT in LACC patients correlates with an increased likelihood of distant failures. Their study raises important concerns about the currently performed weight loss measurements and they suggest to include CT-based muscle measurements into prediction models to improve the prediction of distant failures.

Author Response

For Reviewer #1:

Reviewer #1:

Comment 1: In the submitted study on “Muscle loss after chemoradiotherapy as a biomarker of distant failures in locally advanced cervical cancer” the authors analyzed the potential of SMI and SMD measurements as suitable markers to predict distant failures after CCRT of patients with locally advanced cervical cancer.

The authors systematically and thoroughly analyze BMI, SMI, SMD and TATI for the study cohort and provide comprehensive detail information on their observations supporting the claims made in the manuscript.

The authors discuss their findings critically, considering different aspects of causality and correlation of their observations in the discussion (MS lines 162-180). They explain the overall setup of the study and the selection of the cohort and also point out that the pre-treatment BMI, SMI, SMD and TATI were not significantly different between SMI loss and SMI maintained groups.

Further, they point out the limitations of their study and suggest possible next steps.

Although the authors mention another study analyzing SMI changes in comparison to weight loss measurements (MS line187-189) it does not become sufficiently clear whether this is the same paper they are referring to in Ref 20 (MS line 190) or whether this is another study. Also, the authors do not emphasize the comparison to their study sufficiently, they could point out more clearly that the overall results of both studies are comparable and further strengthen the importance to include SMI measurements in clinical predictions. This should ideally already be mentioned in the introduction where they in line 64 and 65 just refer to the current state of knowledge in the context of LACC. There the authors should refer to their study in a broader context. Are SMI and SMD measurements performed or described to increase predictions in the context of any other form of cancer?

Response 1: We appreciate the reviewer’s valuable suggestions. In line 187-190 of the original manuscript, this was the same reference (Kays et al. J Cachexia Sarcopenia Muscle 2018, 9, 673-684). To improve the manuscript according to the reviewer’s comments, we revised the Introduction and Discussion sections, as follows:

In the Introduction section – (Page 2, lines 64–66)

“Integrating CT-based muscle measurement could improve the prediction of survival outcomes in cancer patients [24,25]; however, the predictive value of muscle loss for distant failures is unknown in LACC.”

In the Discussion section – (Page 4, lines 210–236)

“BMI is not a precise measure of body composition because patients with similar BMI can have different body composition phenotypes [33,34]. Cancer cachexia is characterized by progressive muscle loss with or without loss of fat mass that cannot be fully reversed with nutritional support [35]. However, the current definition of cachexia is based on weight loss without considering the changes in body composition [35]. In a previous study, it was reported that CT-based body composition analysis detected muscle or fat loss of >5% in 81% of patients, while the traditional definition of weight loss of >5% was observed only in 56.6% of patients [21]. If the traditional definition of weight loss of >5% was applied to patients with muscle or fat loss of >5%, this would lead to missing of nearly 30% of these patients as they would have been reported as having developed cachexia in their study. Moreover, three distinct cachexia phenotypes were suggested based on muscle and fat changes and their impact on outcomes in pancreatic cancer patients [21]. In this study, we found that SMI and SMD changes were not correlated with BMI change, and that the TATI change was weakly correlated, although significantly, with BMI change. These findings suggested that the changes in weight during cancer therapy could not have represented muscle change, but might likely represent TATI change. Applying the traditional weight loss criterion to patients with SMI loss would have classified 24/90 patients as having developed cachexia; this would have missed 66/90 or 73.3% of patients with SMI loss. Furthermore, we found that muscle change, but not fat change, was associated with distant failures in LACC patients treated with CCRT. Differences in the effect of fat on outcomes might be owing to differences in cancer types, treatments, or ethnicity. Weight loss based on the current definition of cachexia was not predictive of distant failures in LACC. On predicting distant failures, the muscle-loss model also significantly improved the prediction of the distant failures compared with either the clinical or weight-loss models. Hence, defining cachexia phenotypes based on the CT-based body composition measurement rather than the weight alone, may provide a more precise definition of cachexia phenotypes. Integrating the skeletal muscle measurement into the classical prognostic models could also help improve the prediction of the distant failures in cancer patients.”

Comment 2: Further, I recommend that the authors consider providing more visual display formats for the information provided in the tables and the text. It would be easier for the reader if some comparisons would be plotted as bar graphs in addition to the tables for example in line 143-145 listing the C-indexes.

Response 2: We concur with the reviewer’s pertinent comments. Visual display formats would be easier for the readers to understand our results in this manuscript. We compared only three models in this study and would like to suggest that the comparison of these models be done using texts. To improve our manuscript according to the reviewer’s comments, we revised the Results section as follows:

In the Results section – (Page 2, lines 171–175)

“The C-indexes of the constructed models showed similar indexes between the clinical model (0.756 [95% CI: 0.679–0.832]) and the weight-loss model (0.758 [95% CI: 0.680–0.836]), p<0.06; while that of the muscle-loss model (0.824 [95% CI: 0.748–0.900]) was greater than that of the clinical model (p<0.001) and weight-loss model (p<0.001).”

Comment 3: Same advice holds true for Table 1 and 2, this should be plotted as bar graphs or similar in addition to the data listed in the tables (the tables could still be provided as supplemental information). Table 1 and 2 contain a lot of information and plotting this data is essential in order to present this data in the best comprehensible format.

Response 3: We concur with the reviewer’s comment. To improve the manuscript according to the reviewer’s valuable comments, we revised the Results section, Table 1, Table 2, new Fig. 3, Supplementary Table S1, and Supplementary Table S2 as follows:

 In the Results section – (Pages 3–4, lines 88–113)

“The baseline body composition and changes after CCRT are summarized in Table 2 and Supplementary Table S1. Overall, patients significantly lost body mass index (BMI), skeletal muscle density (SMD), and total adipose tissue index (TATI) after CCRT; the skeletal muscle index (SMI) loss was marginally significant. Fifty-eight (20.9%) patients experienced weight loss of ≥ 5%, while 90 (32.4%), 124 (44.6%), and 125 (45.0%) patients experienced SMI loss, SMD loss, or TATI loss of ≥ 5%, respectively. SMI and SMD changes were not correlated with BMI change (Spearman’s ρ for SMI, 0.11; p=0.06; ρ for SMD, 0.10; p=0.10; Fig. 2A-B). TATI changes were weakly correlated with BMI change (Spearman’s ρ for TATI, 0.35; p<0.001; Fig. 2C). SMI changes were moderately correlated with SMD change (Spearman’s ρ for SMD, 0.54; p<0.001; Fig. 2D). TATI changes were weakly correlated with SMI and SMD changes (Spearman’s ρ for SMI, 0.19; p=0.001; ρ for SMD, 0.12; p=0.06; Fig. 2E-F).

The cut-off values for sarcopenia, myosteatosis, and low TATI were SMI <36.3 cm2/m2, SMD <30.7 HU, and TATI <112.2 cm2/m2, respectively. Pre-treatment BMI, SMI, SMD, and TATI were similar between the SMI loss and SMI maintained groups (Fig. 3; Supplementary Table S1). Twenty-six (28.9%) and 66 (35.1%) patients in the SMI loss and SMI maintained groups, respectively, had pre-treatment sarcopenia (p=0.30). However, 44 (48.9%) and 53 (28.2%) patients in the SMI loss and SMI maintained groups, respectively, had post-treatment sarcopenia (p=0.001). Pre-treatment myosteatosis was present in 31 (34.4%) patients in SMI loss group compared with 61 (32.4%) patients in the SMI maintained group, respectively (p=0.74). Pre-treatment low TATI was present in 57 (63.3%) patients in SMI loss group compared with 129 (68.6%) patients in the SMI maintained group, respectively (p=0.38).

Adenocarcinoma was associated with SMI loss (Table 1). Compared with patients with squamous cell carcinoma (SCC), patients with adenocarcinoma lost more SMI (-5.0% vs. -0.5%; p=0.01) and marginally more SMD (-6.7% vs. -2.4%; p=0.06) (Supplementary Table S2). Demographic characteristics such as FIGO stage, pelvic lymph nodes (PLNs), SCC-antigen (SCC-Ag), and treatments were not significantly different between groups (Table 1).”

In Table 1

Table 1. Patient and tumour characteristics, values expressed as mean ± standard deviation, unless stated otherwise.

Characteristics

Overall

(n = 278)

SMI loss

(n = 90)

SMI maintained

(n = 188)

p-value

Age (years), median (IQR)

62 (53-73)

64 (54-75)

62 (52-71)

0.11

ECOG performance status

0.08

  0

253 (91.0)

78 (86.7)

175 (93.1)

  1

25 (9.0)

12 (13.3)

13 (6.9)

FIGO stage

0.28

  IB-II

203 (73.0)

62 (68.9)

141 (75.0)

  III-IVA

75 (27.0)

28 (31.1)

47 (25.0)

Pathology

0.002

  Squamous cell carcinoma

246 (88.5)

72 (80.0)

174 (92.6)

  Adenocarcinoma

32 (11.5)

18 (20.0)

14 (7.4)

Pelvic lymph node

0.80

  Positive

136 (48.9)

45 (50.0)

91 (48.4)

  Negative

142 (51.1)

45 (50.0)

97 (51.6)

SCC-Ag level, median (IQR)

7.6 (3.3-15.0)

8.6 (2.8-20.4)

6.9 (3.3-14.3)

0.29

Radiation field

0.09

  Extended-field radiotherapy

147 (52.9)

41 (45.6)

106 (56.4)

  Pelvic radiotherapy

131 (47.1)

49 (54.4)

82 (43.6)

Chemotherapy

0.07

  Yes

243 (87.4)

74 (82.2)

169 (89.9)

  No

35 (12.6)

16 (17.8)

19 (10.1)

Chemotherapy cycles

n = 243

n = 74

n = 169

0.78

  5-6

181 (74.5)

56 (75.7)

125 (74.0)

  1-4

62 (25.5)

18 (24.3)

44 (26.0)

Overall treatment duration (day), median (IQR)

58 (54-61)

59 (52-61)

58 (54-61)

0.89

Median (IQR) duration between CT scans, days

143 (135-150)

141 (133-149)

143 (135-151)

0.38

Abbreviations: ECOG, Eastern Cooperative Oncology Group; SCC-Ag, squamous cell carcinoma antigen.

In Table 2

Table 2. Change of body composition parameters after CCRT.

First CT scan

Second CT scan

Relative Change per 150 days (%)

Variable

Mean ± SD

Mean ± SD

Mean

95% CI

p-value

BMI (kg/m2)

23.4 ± 4.3

23.0 ± 4.3

-1.9

-2.4 to -1.4

<0.001

SMI (cm2/m2)

39.8 ± 7.3

39.3 ± 7.6

-1.0

-2.1 to 0.2

0.09

SMD (HU)

35.6 ± 9.5

34.5 ± 9.6

-2.9

-4.4 to -1.5

<0.001

TATI (cm2/m2)

97.7 ± 44.5

93.7 ± 41.8

-3.0

-4.9 to -1.2

0.001

Abbreviations: BMI, body mass index; SMD, skeletal muscle density; SMI, skeletal muscle index; TATI, total adipose tissue index.

In Fig. 3– (Please see the attachment)

Figure 3. Pre-treatment body composition parameters according to skeletal muscle index (SMI) change groups.

In Supplementary Table S1

Supplementary Table S1 Body composition parameters according to SMI change groups, values expressed as mean ± standard deviation, unless stated otherwise.

Characteristics

Overall

(n = 278)

SMI loss

(n = 90)

SMI maintained

(n = 188)

p-value

Pre-treatment BMI (kg/m2)

23.4 ± 4.3

23.4 ± 4.8

23.4 ± 4.1

0.90

Weight loss ≥ 5%, n (%)

58 (20.9)

24 (26.7)

34 (18.1)

0.10

Pre-treatment SMI (cm2/m2)

39.8 ± 7.3

40.6 ± 7.0

39.4 ± 7.4

0.20

Pre-treatment sarcopeniaa, n (%)

92 (33.1)

26 (28.9)

66 (35.1)

0.30

Post-treatment SMI (cm2/m2)

39.3 ± 7.6

36.0 ± 6.5

40.9 ± 7.6

<0.001

  Post-treatment sarcopeniaa, n (%) 

97 (34.9)

44 (48.9)

53 (28.2)

0.001

Pre-treatment SMD (HU)

35.6 ± 9.5

35.6 ± 9.2

35.6 ± 9.6

0.84

  Pre-treatment myosteatosisa, n (%)

92 (33.1)

31 (34.4)

61 (32.4)

0.74

SMD change (%/150 days)

-2.9 ± 12.4

-11.2 ± 10.0

1.0 ± 11.5

<0.001

  SMD maintained

154 (55.4)

23 (25.6)

131 (69.7)

<0.001

  SMD loss ≥ 5%

124 (44.6)

67 (74.4)

57 (30.3)

Pre-treatment TATI (cm2/m2)

97.7 ± 44.5

96.9 ± 47.8

98.0 ± 42.9

0.62

Pre-treatment low TATIa, n (%)

186 (66.9)

57 (63.3)

129 (68.6)

0.38

TATI change (%/150 days)

-3.0 ± 15.7

-5.8 ± 16.5

-1.7 ± 15.2

0.04

  TATI maintained

153 (55.0)

41 (45.6)

112 (59.6)

0.03

  TATI loss ≥ 5%

125 (45.0)

49 (54.4)

76 (40.4)

Abbreviations: BMI, body mass index; HU, Hounsfield unit; IQR, interquartile range; SMD, skeletal muscle density; SMI, skeletal muscle index; TATI, total adipose tissue index.

a SMI < 36.3 cm2/m2, SMD < 30.7 HU, and TATI < 112.2 cm2/m2 were defined as sarcopenia, myosteatosis, and low TATI, respectively.

In Supplementary Table S2

Supplementary Table S2 Change of body composition parameters during treatment.

First CT scan

Second CT scan

Relative Change per 150 days (%)

Variable

Mean ± SD

Mean ± SD

Mean

95% CI

p-value

BMI (kg/m2)

  SCC

23.3 ± 4.4

22.9 ± 4.3

-1.9

-2.5 to -1.4

<0.001

  Adenocarcinoma

24.1 ± 3.9

23.6 ± 3.7

-2.0

-3.6 to -0.3

0.03

SMI (cm2/m2)

  SCC

39.9 ± 7.3

39.6 ± 7.5

-0.5

-1.6 to 0.7

0.45

  Adenocarcinoma

39.0 ± 7.2

37.3 ± 8.1

-5.0

-9.1 to -0.9

0.02

SMD (HU)

  SCC

35.6 ± 9.5

34.6 ± 9.6

-2.4

-4.0 to -0.9

0.002

  Adenocarcinoma

35.8 ± 9.2

33.5 ± 9.4

-6.7

-11.3 to -2.2

0.005

TATI (cm2/m2)

  SCC

97.7 ± 45.3

93.7 ± 42.8

-3.2

-5.2 to -1.1

0.002

  Adenocarcinoma

97.1 ± 38.3

93.7 ± 33.8

-2.1

-7.0 to 2.7

0.38

SCC, squamous cell carcinoma.

Comment 4: Overall, the authors could convincingly show that muscle loss after CCRT in LACC patients correlates with an increased likelihood of distant failures. Their study raises important concerns about the currently performed weight loss measurements and they suggest to include CT-based muscle measurements into prediction models to improve the prediction of distant failures.

Response 4: We appreciate the reviewer’s time and effort in reviewing this manuscript. Although this study has these strengths, it also has major weaknesses and limitations. We have thoroughly revised this manuscript according to the reviewer’s comments. The suggested changes have enriched the manuscript and produced a more balanced and improved account of the research.

Reviewer 2 Report

Dear Authors,

The article “Muscle loss after chemoradiotherapy as a biomarker of distant failures in locally advanced cervical cancer” is very interesting work, reporting the first study connected with predictive evaluation of CT-based skeletal muscle measurement as a patient-specific biomarker of distant failure in LACC. This retrospective study reports the correlation between muscle loss after CCRT and probability of distant failures. It shows, that integration of CT-based muscle measurement into classical prognostic factors models may significantly improve the prediction for distant failures in LACC, as the weight loss based on current definition of cachexia can’t be considered as a good predictor in abovementioned model.

The article is well-written with proper study design and statistical analysis. Moreover, it may positively influence the cancer therapy outcome prediction and serve as supportive tool in the prediction of distant failures in locally advanced cancers.

English is good, although I would encourage a careful check of the article to seek potential typos and minor mistakes. I recommend that the article is accepted in present form, but prior to publication it would be good if Authors enlarge the figures 2 and 3, as not all captions can be read easily.

Kind regards

Author Response

For Reviewer #2:

Reviewer #2:

Comment 1: The article “Muscle loss after chemoradiotherapy as a biomarker of distant failures in locally advanced cervical cancer” is very interesting work, reporting the first study connected with predictive evaluation of CT-based skeletal muscle measurement as a patient-specific biomarker of distant failure in LACC. This retrospective study reports the correlation between muscle loss after CCRT and probability of distant failures. It shows, that integration of CT-based muscle measurement into classical prognostic factors models may significantly improve the prediction for distant failures in LACC, as the weight loss based on current definition of cachexia can’t be considered as a good predictor in abovementioned model.

The article is well-written with proper study design and statistical analysis. Moreover, it may positively influence the cancer therapy outcome prediction and serve as supportive tool in the prediction of distant failures in locally advanced cancers.

English is good, although I would encourage a careful check of the article to seek potential typos and minor mistakes. I recommend that the article is accepted in present form, but prior to publication it would be good if Authors enlarge the figures 2 and 3, as not all captions can be read easily.

Response 1: We appreciate the reviewer’s time and effort in reviewing this manuscript. To improve the original Figure 2 and Figure 3 according to the reviewer’s comment, we revised these figures in the attachment.

Reviewer 3 Report

In the present manuscript, authors have demonstrated that muscle loss after chemo-radiotherapy was independently associated with distant failures. They have analyzed the data with different permutation and combination along with BMI, fat mass, muscle index, etc. and conclude that muscle loss is an independent parameter associated with distant failure in colorectal cancer. The overall study is very significant and provides important insight for making clinical decisions in the patients undergoing muscle loos during therapy. It will be really good if the authors can provide a survival curve between combine muscle and fat loss verses no muscle and fat loss in Figure 3. Also, it will be good if in figure 2 they can present a correlation between TATI and SMI/SMD. Adding statistical analysis description in figure legends will be good.

Author Response

For Reviewer #3:

Reviewer #3:

Comment 1: In the present manuscript, authors have demonstrated that muscle loss after chemo-radiotherapy was independently associated with distant failures. They have analyzed the data with different permutation and combination along with BMI, fat mass, muscle index, etc. and conclude that muscle loss is an independent parameter associated with distant failure in colorectal cancer. The overall study is very significant and provides important insight for making clinical decisions in the patients undergoing muscle loos during therapy. It will be really good if the authors can provide a survival curve between combine muscle and fat loss verses no muscle and fat loss in Figure 3. Also, it will be good if in figure 2 they can present a correlation between TATI and SMI/SMD. Adding statistical analysis description in figure legends will be good.

Response 1: We appreciate the reviewer’s time and effort in reviewing this manuscript. We revised the Results section and created new survival curves demonstrating the survival curve of the combined muscle and fat losses versus no muscle or fat losses, as well as the scatter plots presenting the correlation between TATI and SMI/SMD in Figure 2 and Figure 5 as follows:

In the Results section – (Pages 3–4, lines 96–98)

“TATI changes were weakly correlated with SMI and SMD changes (Spearman’s ρ for SMI, 0.19; p=0.001; ρ for SMD, 0.12; p=0.06; Fig. 2E-F).”

In the Results section – (Page 6, lines 143–145)

“Stratifying the patients into three categories: “No SMI loss”, “Only SMI loss”, and “SMI and TATI losses”; the 3-year DRFS were 93.3%, 52.6%, and 59.2%, respectively (p<0.001; Fig. 5).”

In the Figure 2 and Figure 5: (Please see the attachment)
